# New-Onset Paroxysmal Atrial Fibrillation in the Setting of Acute Pulmonary Embolism Is Associated with All-Cause Hospital Mortality in Women but Not in Men

**DOI:** 10.3390/diagnostics13111829

**Published:** 2023-05-23

**Authors:** Ivica Djuric, Boris Dzudovic, Bojana Subotic, Jelena Dzudovic, Jovan Matijasevic, Marija Benic, Sonja Salinger, Irena Mitevska, Ljiljana Kos, Tamara Kovacevic-Preradovic, Stefan Simovic, Vladimir Miloradovic, Tanja Savicic, Bjanka Bozovic, Nebojsa Bulatovic, Srdjan Kafedzic, Aleksandar N. Neskovic, Nikola Kocev, Jelena Marinković, Slobodan Obradovic

**Affiliations:** 1Clinic of Cardiology, Military Medical Academy, 11000 Belgrade, Serbiasloba.d.obradovic@gmail.com (S.O.); 2Clinic of Emergency Internal Medicine, Military Medical Academy, 11000 Belgrade, Serbia; dzuda1977@gmail.com; 3School of Medicine Military Medical Academy, University of Defense, 11000 Belgrade, Serbia; 4National Poison Control Center, Military Medical Academy, 11000 Belgrade, Serbia; 5Institute for Pulmonary Diseases of Vojvodina, 21204 Sremska Kamenica, Serbia; 6School of Medicine, University of Novi Sad, 21000 Novi Sad, Serbia; 7Clinic of Cardiology, Clinical Center Nis, University of Nis, 18000 Nis, Serbia; 8Clinic of Cardiology, School of Medicine, University of Skopje, 1000 Skopje, North Macedonia; 9Clinic of Cardiology, Clinical Center Banja Luka, School of Medicine, University of Banja Luka, 78000 Banja Luka, Bosnia and Herzegovina; 10Clinic of Cardiology, Clinical Center Kragujevac, School of Medicine, University of Kragujevac, 34000 Kragujevac, Serbia; 11Department for Internal Medicine, General Hospital Pancevo, 26000 Pancevo, Serbia; 12Clinic of Cardiology, Clinical Center Podgorica, 81000 Podgorica, Montenegro; 13School of Medicine Podgorica, University of Podgorica, 81000 Podgorica, Montenegro; 14Department of Cardiology, Clinical Hospital Center Zemun, 11080 Zemun, Serbia; skafedzic@yahoo.com (S.K.);; 15School of Medicine Belgrade, University of Belgrade, 11000 Beograd, Serbia; 16Institute for Medical Statistics School of Medicine, University of Belgrade, 11000 Beograd, Serbia

**Keywords:** paroxysmal atrial fibrillation, pulmonary embolism, mortality risk

## Abstract

Background: Patients with acute pulmonary embolism (PE) may have various types of atrial fibrillation (AF). The role of AF in hemodynamic states and outcomes may differ between men and women. Methods: In total, 1600 patients (743 males and 857 females) with acute PE were enrolled in this study. The severity of PE was assessed using the European Society of Cardiology (ESC) mortality risk model. Patients were allocated into three groups according to their electrocardiography recordings taken during hospitalization: sinus rhythm, new-onset paroxysmal AF, and persistent/permanent AF. The association between the types of AF and all-cause hospital mortality was tested using regression models and net reclassification index (NRI) and integrated discrimination index (IDI) statistics with respect to sex. Results: There were no differences between the frequencies of the types of AF between men and women: 8.1% vs. 9.1% and 7.5% vs. 7.5% (*p* = 0.766) for paroxysmal and persistent/permanent AF, respectively. We found that the rates of paroxysmal AF significantly increased across the mortality risk strata in both sexes. Among the types of AF, the presence of paroxysmal AF had a predictive value for all-cause hospital mortality independent of mortality risk and age in women only (adjusted HR, 2.072; 95% CI, 1.274–3.371; *p* = 0.003). Adding paroxysmal AF to the ESC risk model did not improve the reclassification of patient risk for the prediction of all-cause mortality, but instead enhanced the discriminative power of the existing model in women only (NRI, not significant; IDI, 0.022 (95% CI, 0.004–0.063); *p* = 0.013). Conclusion: The occurrence of paroxysmal AF in female patients with acute PE has predictive value for all-cause hospital mortality independent of age and mortality risk.

## 1. Introduction

Atrial fibrillation (AF) has a broad presence in various chronic and acute diseases. Its occurrence is associated with adverse prognosis, and in some acute settings, it can directly compromise a patient’s hemodynamics and contribute to a lethal outcome. There is a high incidence of atrial fibrillation in patients with acute pulmonary embolism and this percentage varies between 12 and 24%, according to most recent studies [1,2,3,4]. Patients may have paroxysmal (pAF) or persistent/permanent AF, which can be a consequence of chronic heart disease; together, AF and chronic heart disease can diminish cardio-pulmonary reserves, worsen adaptation to acute pulmonary flow obstruction, and contribute to hemodynamic collapse. According to recent studies, there are significant gender differences in AF patients, especially in terms of age, comorbidities, the risk of stroke, and the presence of heart failure with preserved systolic function [5,6]. The function of the right ventricle (RV) differs in some important details in men compared to women. Healthy men have higher RV mass, stroke volume, end-diastolic volumes, and end-systolic volumes, while women have a higher RV ejection fraction [7]. These differences can be very important in some acute pathophysiological circumstances with sudden RV overload. For instance, female patients with inferior ST-elevation myocardial infarctions more often have signs of RV involvement and dysfunction compared to male patients [8]. In acute pulmonary embolism, extensive obstruction of the pulmonary arteries overwhelms the thin RV, expanding RV volume and provoking significant RV dysfunction. The weaker RV wall in females could be more prone to the development of pAF and the occurrence of pAF could be more deleterious in women compared to men. There are several biomarkers that we can use as indicators of right ventricular dysfunction and predictors of clinical outcome in pulmonary embolism and atrial fibrillation. Brain natriuretic peptide (BNP) is synthesized as a prohormone (proBNP) in cardiomyocytes, after which it is released into circulation as a reaction to right ventricle stress. After release into circulation, proBNP is cleaved into BNP and an N-terminal fragment (NT-proBNP) in equimolar proportions. The plasma levels of natriuretic peptides reflect the severity of RV dysfunction and hemodynamic compromise in acute PE and play a role in the risk stratification of patients with PE [9,10]. In patients with AF, the blood levels of BNP increase during AF onset and decrease with rhythm restoration [11,12]. Important gender differences exist regarding BNP blood levels in patients with AF, and women have been found to have higher median plasma BNP levels than men, independent of cardiovascular diseases or dysfunction [13].

The presence of AF has been associated with higher hospital mortality in several retrospective studies, and 30-day mortality varies between 6.5% and 30.1% and between 19.4% and 25% for paroxysmal and persistent/permanent AF, respectively [1,14,15,16,17,18,19]. Since there are insufficient data in the literature considering the frequency and consequences of the occurrence of pAF in acute PE patients regarding sex, we aimed to determine the impact of AF type (pAF compared to persistent/permanent AF) on sex-related short-term all-cause mortality in patients with acute pulmonary embolism.

## 2. Materials and Methods

The present retrospective cross-sectional study included patients with acute PE from the Regional Pulmonary Embolism Register (REPER), in which 8 hospitals were enrolled in the period from 2014 to 2021. In total, 1600 patients were included in the study from the 1612 enrolled in the register. Twelve patients were excluded due to having rhythms other than sinus rhythm or atrial fibrillation. All patients were diagnosed with PE using multi-detector-computed tomography pulmonary angiography (MDCT-PA) and treated according to the current guidelines for PE. A standard 12-lead surface electrocardiogram (ECG) was collected upon hospital admission. For patients with low and intermediately low risk, an ECG was recorded every 24 h during hospitalization, with more frequent recordings if clinically indicated. For patients with intermediately high to high risk, a 12-channel ECG was obtained every 12 h, and these patients were monitored for the duration of their hospitalization. Patients were allocated into the men or women groups according to sex, and then into three groups: (1) the sinus rhythm group, which included patients with sinus rhythm at admission that lasted the duration of their hospitalization; (2) the paroxysmal AF group, which included patients who developed one or more episodes of new-onset AF with a duration more than 30 s and less than 7 days across the duration of their hospitalization; and (3) the persistent/permanent AF group, which included patients who had ever had AF lasting more than 7 days. The patient demographic and clinical characteristics, which included symptoms, comorbidities, laboratory findings, and hemodynamic parameters, were collected and analyzed. For each patient, sPESI (simplified Pulmonary Embolism Severity Index) and creatinine clearance values were calculated (Cockcroft–Gault equation). The severity of PE was assessed using the European Society of Cardiology mortality risk model. Brain natriuretic peptide (BNP) plasma level assays (fully automated two-site sandwich chemiluminescent immunoassays for the measurement of BNP in EDTA plasma samples) were conducted using patient blood samples taken during the first 24 h after admission (Siemens Healthcare Diagnostics). Serum concentrations of cTnI were quantified using a conventional ADVIA Centaur ultra-cTnI assay (Siemens Healthcare Diagnostics), a fully automated three-site sandwich immunoassay using direct chemiluminometric technology. The primary end-point of the study was all-cause hospital mortality, and the secondary end-point of the study was whether adding paroxysmal AF to a mortality risk model for female patients could significantly increase its prognostic power.

Age is presented as means and standard deviations, and other patient characteristics are presented as frequencies. The differences in age related to AF type were calculated using ANOVA, and differences between categorical variables were tested using the Chi-square test. The association between patient characteristics and all-cause mortality was estimated with a univariate and multivariable Cox regression analysis, where variables with *p* < 0.05 in the univariate analysis were entered into a multivariable Cox regression model. *p*-values less than 0.05 were considered significant. BNP and cTnI blood levels were compared with an independently sampled Mann–Whitney U test, and the biomarker levels are presented as medians and 25–75th percentile values. The REPER database was accessed, and all statistical analyses were performed with BM SPSS Statistics for Windows, Version 20.0. Armonk, NY: IBM Corp. The integrated discrimination index (IDI), net reclassification index (NRI), and median improvement in risk score (MIRS) values were calculated using the latest version of the survIDINRI package, version 1.1–2, released in April 2022. The Cox regression proportional hazard model for ESC mortality risk stratification was used as the working model for the prediction of all-cause and PE-related hospital mortality. After adding paroxysmal AF with respect to sex to the ESC risk stratification model, improvement in the prediction of all-cause and PE-related hospital mortality was recorded in the form of IDI, NRI, and MIRS values, as well as their associated 95% confidence intervals and *p*-values [20].

## 3. Results

In the REPER, 1612 patients with acute PE were enrolled, and we identified 258 (16.0%) patients with AF. Among them, 124 (7.69%) patients had new-onset paroxysmal AF, while 108 (6.71%) had persistent/permanent AF. The flowchart of this study is presented in Figure 1.

Regarding AF types in men vs. women, the prevalence values were 8.1% vs. 9.1% and 7.5% vs. 7.5% (*p*-values not significant) for new-onset paroxysmal AF and persistent/permanent AF, respectively. The characteristics of patients regarding sex and type of AF are listed in Table 1. The patients with AF were older than the patients presenting with sinus rhythm, and the patients with permanent atrial fibrillation were the oldest. In the female cohort, the maximum age difference between patients was more than 10 years. There were no statistically significant differences in the presence of chronic obstructive pulmonary disease, arterial hypertension, major surgery in the past 6 months, a glomerular filtration rate (GFR) less than 30 mL/min, and right ventricle dysfunction in either the sex-related groups or the AF-type-related groups. Concerning coronary artery disease in the male cohort, AF-type-related differences were statistically significant, but in the female cohort, there were no differences between the patients with sinus rhythm and those with both types of AF. For male patients, the presence of diabetes and prior stroke was significantly different between AF groups, but this was not the case among female patients. When data existed on the presence of congestive heart failure, either paroxysmal or permanent/persistent AF was significantly present in both sexes. As can be seen in Table 1, the proportion of patients with new-onset paroxysmal AF increases with higher PE mortality risk in both sexes.

We analyzed several hemodynamic parameters such as heart rate, systolic arterial pressure, and O_2_ saturation, but found no statistically significant differences on the basis of sex. Additionally, when comparing patients with paroxysmal AF and acute PE on the basis of sex, we did not find significant differences in several echocardiographic measures, such as right ventricle diameter (RV diameter), right ventricle systolic pressure (RVSP), tricuspid annular plane systolic excursion (TAPSE), TAPSE/RSPV index, and RV/(left ventricle) LV index. Regarding blood analysis (hemoglobin levels, hematocrit levels, and total leucocyte count) and biomarkers (C-reactive protein, brain natriuretic peptide (BNP), and cardiac troponin I (cTnI)), we also did not find significant differences on the basis of sex, except in the case of the BNP analysis of a cohort of patients with left ventricle ejection fractions higher than 50%; this result is shown further down (Appendix A). New-onset paroxysmal AF, particularly in women, was associated with significantly higher hospital mortality in intermediately high-risk and high-risk PE patients compared to those without new-onset paroxysmal AF (Appendix A). The all-cause hospital mortality values were 8.0% vs. 20.4% vs. 21.4% in men and 9.0% vs. 30.8% vs. 15.6% in women for sinus rhythm, paroxysmal AF, and persistent/permanent AF, respectively. A Kaplan–Meier curve analysis found that the survival rate was significantly lower for both types of AF in men (log rank, *p* = 0.005), but only for paroxysmal AF in women (log rank, *p* < 0.001) (Figure 2).

The unadjusted hazard ratio for all-cause hospital mortality shows that new-onset paroxysmal AF indicated significant risk [HR, 2.66; 95% confidence interval (CI), 1.41–4.99; and *p* = 0.002 for men and HR, 3.82; 95% confidence interval (CI), 2.39–6.11; and *p* < 0.001 for women]. Persistent/permanent AF only represented higher risk for hospital mortality in men [HR, 2.82; 95% confidence interval (CI), 1.50–5.30; and *p* < 0.001]. After we performed multivariate regression analysis adjusted for age and mortality risk, only the presence of new-onset paroxysmal atrial fibrillation in women had a predictive value for all-cause hospital mortality [HR, 2.072; 95% confidence interval (CI), 1.27–3.37; and *p* = 0.003] (Table 2).

Using integrated discrimination index (IDI) and net reclassification index (NRI) statistics, (Table 3), we found that adding a new-onset paroxysmal AF variable to the ESC mortality risk model using Cox regression did not improve the reclassification of patients in any gender group with respect to the prediction of all-cause and PE-related hospital mortality (NRI was not significant for both outcomes). However, in women, adding the new-onset paroxysmal AF variable to the ESC mortality risk model significantly improved its prognostic power for both all-cause and PE-related hospital mortality (IDI was significant for both outcomes) inside of the pre-defined risk strata (Table 3).

In order to explain the different prognostic significance of paroxysmal AF in women compared to men, we compared serum levels of BNP measured within the first 24 h after admission. Patients with left ventricle ejection fractions lower than 50% were excluded from this analysis. Serum BNP levels were significantly higher in women compared to men, with 388 pg/mL (25–75th percentile, 246–579 pg/mL) vs. 185 pg/mL (25–75th percentile, 121–364 pg/mL), respectively, *p* = 0.045 (Figure 3). On the other hand, in patients with paroxysmal AF, there was no difference in serum cTnI levels at admission in women compared to men, with 0.12 ng/mL (25–75th percentile, 0,028–0.340 ng/mL) vs. 0.070 ng/mL (25–75th percentile, 0.022–0.575 ng/mL), respectively, *p* = ns. There was also no significant difference in glomerular filtration rate (calculated using the Cocroft formula with creatinine serum levels at admission) between women and men, with paroxysmal AF 53 mL/min (25–75th percentile, 34–72 mL/min) vs. 63 mL/min (25–75th percentile, 50–77 mL/min), respectively, *p* = 0.07.

## 4. Discussion

The main finding of our study is that new-onset paroxysmal AF in women with acute PE is a significant predictor of hospital mortality, independent of ESC mortality risk and age. On the contrary, in men, new-onset paroxysmal AF during acute PE does not have independently predictive value for hospital mortality (adjusted for mortality risk and age). Additionally, persistent/permanent AF does not have an independently predictive value for hospital mortality in both men and women.

There have only been a few published studies that focused on different types of AF in patients with PE, but none of them, according to our knowledge, analyzed patients according to sex [3,4,21,22]. In addition, the results of recent studies regarding the impact of AF on early PE-related hospital mortality are not consistent [1,14,15,16,17,18,19]. The incidence of atrial fibrillation in the REPER is similar to that of other registries, around 16.12%, and our distribution of the different types of atrial fibrillation is also similar to the distributions of other studies. In a study by Krajewska et al., the incidence of atrial fibrillation was almost the same, 16%, and the incidence of paroxysmal AF was slightly lower, 7.9% [18]. In a study by Liu D. et al., who defined new-onset atrial fibrillation in a similar way to Krajewska et al., the incidence of new-onset AF was 7.16%, similar to our results [21]. To the best of our knowledge, the only data for sex distribution on this topic are from a study by Ng A.C.C. et al., who found a slightly higher incidence in men than in women, 1167 per 100,000 person-years vs. 840 per 100,000 person-years, which is not consistent with our data [19]. According to Keller K et al., the all-cause hospital mortality of PE patients decreased from 20.4% to 13.9% in the last decade [22]. The explanation for this may be related to improvements in risk factor identification, better treatment protocols, and the improved sensitivity of diagnostic modalities [22]. There are insufficient available data on sex-related differences in all-cause hospital mortality, but according to data from a study by Agarwal S. et al., women with acute PE had significantly higher all-cause hospital mortality compared to men (odds ratio OR, 1.09; 95% confidence interval CI, 1.03 to 1.015) [23]. In a study by Barra et al., the all-cause hospital mortality of patients with PE and AF was 22.8% and the 1-month mortality was 35.1%, which is consistent with our data [1]. However, most authors have not distinguished between patients according to sex and some of them have not even distinguished according to types of AF, making it difficult to draw comparisons. The higher mortality rate in women with new-onset paroxysmal fibrillation and acute PE is probably due to multiple reasons. Persistent AF with fast ventricle response can diminish cardio-pulmonary reserve and contribute to hemodynamic collapse. In the same manner, the loss of the corresponding right ventricle contribution to the preloading of the left ventricle in AF can aggravate hemodynamic disturbances in patients with PE [24,25]. In particular, in patients with permanent AF, the thrombotic embolus may originate from the right atrium [26,27]. AF promotes a pro-thrombotic state by inducing the production of platelets and coagulation cascades, which can lead to thrombosis and eventual PE. On the contrary, severe acute pulmonary artery obstruction can cause AF, most likely due to severe stretching of the right heart wall [24]. As a consequence of this right heart wall stretching, the blood levels of BNP and NT-pro-BNP are raised. According to previous studies, important gender differences exist regarding BNP blood levels, and female patients have higher median plasma BNP levels than male patients [13]. In our study population, we found a similar result, with the considerable difference that serum BNP levels were significantly higher in women in our study compared with data from other studies. In this cohort of patients, we excluded patients with left ventricle ejection fractions lower than 50% to exclude the influence of left ventricle dysfunction. Due to a lack of data, we did not exclude the influence of left atrial cardiomyopathy on BNP levels, choosing instead to analyze it because of its connection to AF. We excluded the influence of other possible reasons for high levels of BNP in the female sex, primarily non-cardiac factors such as low hemoglobin and hematocrit (Appendix A). However, the effects of female sex hormones could be a topic for discussion, as we did not have a sufficient amount of data to analyze [6]. The causes and consequences of AF in PE are still a matter of debate. Our results show that adding the AF variable to ESC mortality risk stratification did not lead to the reclassification of patients to higher-risk groups, which is consistent with the findings in other studies [28]. Interestingly, in women, adding a new-onset paroxysmal AF variable to the ESC mortality risk model did significantly improve its prognostic power for both all-cause and PE-related hospital mortality. In clinical practice, this fact could be helpful in decision-making when therapy escalation is an option. There are several limitations of our study. Similar to most retrospective analyses from multi-centric registers, some data are missing; however, less than 2.0% of values are missing for all relevant variables in our study. Since there was no continuous recording of ECGs, some episodes of AF might be missing; however, the recorded incidence of paroxysmal AF during acute PE is similar to those of other studies, and we are convinced that almost all clinically relevant episodes of AF were detected. Regarding all-cause hospital mortality as the primary outcome of our study instead of PE-related mortality, we chose the most certain end-point, which also represents a mixture of causes where the participation of PE is at least contributory. This is because we wanted to avoid different interpretations of PE-related mortality from different doctors and hospitals. The number of patients in this study was probably not sufficient to investigate all the associations between AF types, sex, and early mortality rates in acute PE. However, it was sufficient for the valid estimation of the predictive value of new-onset paroxysmal AF in women with acute PE.

## 5. Conclusions

New-onset paroxysmal atrial fibrillation in the setting of acute pulmonary embolism was found to be an independent predictor of all-cause hospital mortality in women, but not in men. Adding paroxysmal AF to a mortality risk model for female patients can significantly increase its prognostic power.

## Figures and Tables

**Figure 1 diagnostics-13-01829-f001:**
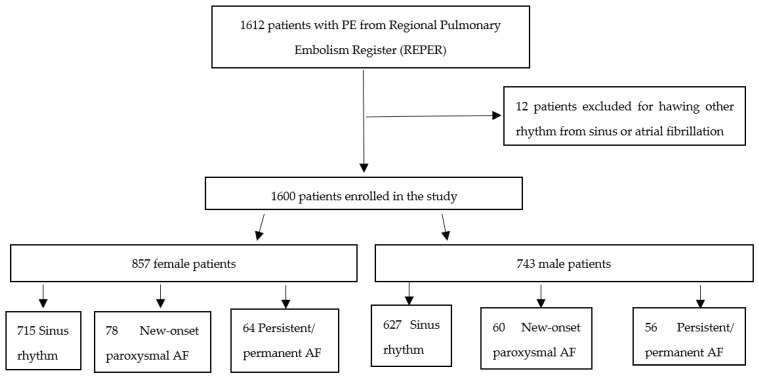
Flowchart of this study.

**Figure 2 diagnostics-13-01829-f002:**
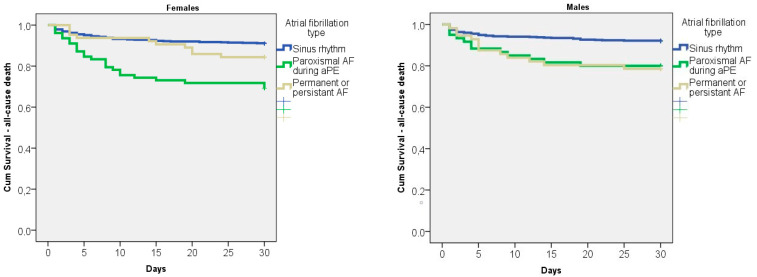
Kaplan–Meier curve of hospital survival with respect to type of AF in females and males.

**Figure 3 diagnostics-13-01829-f003:**
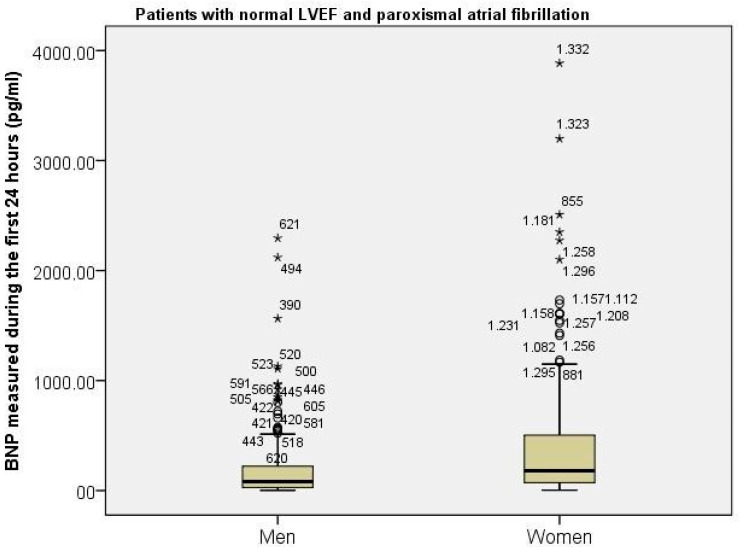
Serum BNP levels measured during the first 24 h after admission between sexes in patients with normal left ventricle ejection fractions.

**Table 1 diagnostics-13-01829-t001:** Baseline characteristics of patients regarding presence of sinus rhythm or different types of atrial fibrillation.

Characteristic Mean ± SD or N (%) According to Presence of AF	Males (N = 743)				Females (N = 857)			
	Controls (Sinus Rhythm) N = 627	Paroxysmal AF N = 60	Persistent/Permanent AF N = 56	*p*	Controls (Sinus Rhythm) N = 715	Paroxysmal AF N = 78	Persistent/Permanent AF N = 64	*p*
**Demographic data**								
Age in years Mean ± SD	58 ± 16	68 ± 10	72 ± 10	<0.001	65 ± 15	73 ± 12	76 ± 9	<0.001
**Medical history**								
COPD, no. (%)	68 (10.8)	3 (5.0)	12 (21.4)	0.016	69 (9.7)	8 (10.3)	8 (12.5)	0.762
Missing data, no. (%)	7 (1.0)				5 (0.7)			
Coronary artery disease	67 (10.8)	9 (15.0)	15 (26.8)	0.002	74 (10.4)	13 (16.9)	6 (9.7)	0.208
Missing data, no. (%)	11 (1.6)				9 (1.2)			
Heart failure	68 (10.8)	17 (28.3)	30 (53.6)	<0.001	81 (11.3)	17 (21.8)	22 (34.4)	<0.001
Missing data, no. (%)	7 (0.9)				5 (0.6)			
Prior stroke	29 (4.6)	10 (16.7)	11 (19.6)	<0.001	49 (6.9)	10 (12.8)	6 (9.4)	0.143
Missing data, no. (%)	9 (1.2)				5 (0.6)			
Diabetes mellitus	97 (15.5)	19 (31.7)	15 (26.8)	0.001	147 (20.6)	22 (28.2)	20 (31.2)	0.055
Missing data, no. (%)	8 (1.1)				5 (0.6)			
Arterial hypertension	322 (51.7)	39 (65.0)	38 (67.9)	0.014	470 (65.8)	54 (69.2)	46 (74.2)	0.361
Missing data, no. (%)	11 (1.5)				8 (0.9)			
Surgery within 6 months	69 (11.0)	8 (13.3)	7 (12.5)	0.826	129 (18.0)	12 (15.4)	15 (23.4)	0.448
Missing data, no. (%)	7 (0.9)				5 (0.6)			
Cancer in last 6 months	78 (12.4)	10 (16.7)	5 (8.9)	0.448	100 (14.0)	11 (14.1)	10 (15.6)	0.937
Missing data, no. (%)	7 (0.9)				5 (0.6)			
RV dysfunction	325 (56.2)	36 (67.9)	39 (75.0)	0.011	402 (61.7)	53 (71.6)	37 (66.1)	0.214
Missing data, no. (%)	67 (8.9)				80 (9.3)			
**Renal failure**								
GFR ˂ 60 mL/min	167 (26.8)	27 (45.8)	25 (44.6)	<0.001	260 (36.7)	42 (53.8)	40 (63.5)	<0.001
Missing data, no. (%)	12 (1.6)				12 (1.4)			
GFR ˂ 30 mL/min	31 (5.0)	5 (8.3)	7 (12.7)	0.043	68 (9.6)	12 (15.4)	13 (20.6)	0.012
Missing data, no. (%)	12 (1.8)				15 (1.7)			
**PE severity**								
Low risk	218 (34.8)	11 (18.3)	12 (21.4)		253 (35.4)	12 (15.4)	16 (25.0)	
Intermediate low risk	162 (25.8)	11 (18.3)	9 (16.1)		167 (23.4)	14 (17.9)	11 (17.2)	
				<0.001				<0.001
High risk	62 (9.9)	12 (20.0)	12 (21.4)		80 (11.2)	24 (30.8)	9 (14.1)	
Missing data, no. (%)	7 (0.9)				5 (0.6)			

AF: atrial fibrillation; SD: standard deviation; COPD: chronic obstructive pulmonary disease; RV: right ventricle, GFR: glomerular filtration rate; PE: pulmonary embolism.

**Table 2 diagnostics-13-01829-t002:** Predictors of all-cause hospital mortality in patients with AF in acute PE in univariate and multivariate models.

**Men**				
	Univariate regression analysis		Multivariate regression analysis	
	HR (95% CI) for all-cause mortality	*p*	HR (95% CI) for all-cause mortality	*p*
Age	1.027 (1.010–1.043)	0.001	1.017 (1.000–1.036)	0.055
Paroxysmal AF	2.656 (1.414–4.988)	0.002	1.610 (0.847–3.061)	0.146
Permanent AF	2.821 (1.502–5.298)	0.001	1.534 (0.790–2.978)	0.206
Intermediately low risk	1.322 (0.496–3.523)	0.577	1.092 (0.395–3.015)	0.865
Intermediately high risk	3.891 (1.779–8.512)	0.001	3.311 (1.503–7.293)	0.030
High risk	13.285 (6.103–28.921)	<0.001	10.978 (4.985–24.176)	<0.001
**Women**				
	Univariate regression analysis		Multivariate regression analysis	
	HR (95% CI) for all-cause mortality	*p*	HR (95% CI) all-cause mortality	*p*
Age	1.035 (1.017–1.052)	<0.001	1.029 (1.011–1.048)	0.002
Paroxysmal AF	3.817 (2.387–6.105)	<0.001	2.072 (1.274–3.371)	0.003
Permanent AF	1.757 (0.902–3.422)	0.097	1.079 (1.011–1.048)	0.828
Intermediately low risk	4.520 (1.643–12.437)	0.003	3.982 (1.446–10.968)	0.008
Intermediately high risk	9.105 (3.598–23.043)	<0.001	7.413 (2.918–18.831)	<0.001
High risk	23.387 (9.202–59.436)	<0.001	18.461 (7.198–47.345)	<0.001

HR: hazard ratio; CI: confidence interval; AF: atrial fibrillation.

**Table 3 diagnostics-13-01829-t003:** Integrated discrimination index (IDI), net reclassification index (NRI), and median improvement in risk score (MIRS) when the presence of paroxysmal AF was added as a variable to the ESC mortality risk model for acute PE in the prediction of all-cause and PE-related hospital mortality.

Outcomes	Men	Women
All-cause mortality		
IDI	0.005 (95% CI, −0.002–0.035), *p* = 0.286	0.022 (95% CI, 0.004–0.063), *p* = 0.013
NRI	0.120 (95% CI, −0.205–0.283), *p* = 0.306	0.180 (95% CI, −0.201–0.275), *p* = 0.525
MIRS	−0.007 (95% CI, −0.016–0.002), *p* = 0.100	−0.016 (95% CI, −0.024–0.000), *p* = 0.000
PE-related mortality		
IDI	0.005 (95% CI, −0.005–0.054), *p* = 0.379	0.010 (95% CI, 0.000–0.041), *p* = 0.040
NRI	0.134 (95% CI, −0.471–0.450), *p* = 0.571	0.165 (95% CI, −0.232–0.257), *p* = 0.651
MIRS	−0.004 (95% CI, −0.024–0.007), *p* = 0.213	−0.007 (95% CI, −0.014–−0.001), *p* = 0.013

AF: atrial fibrillation, PE: pulmonary embolism, ESC: European Society of Cardiology, IDI: integrated discrimination index, NRI: net reclassification index, MIRS: median improvement in risk score.

## Data Availability

The data presented in this study are available on request from the corresponding author.

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
