# Peer review of "New-Onset Paroxysmal Atrial Fibrillation in the Setting of Acute Pulmonary Embolism Is Associated with All-Cause Hospital Mortality in Women but Not in Men"

_diagnostics, 2023, doi:10.3390/diagnostics13111829_

Round 1

Reviewer 1 Report

In this work , Djuric et al highlight the value of new onset atrial fibrillation in acute PE in all cause hospital death in women but not in men.

These findings are well demonstrated, documented, illustrated and coud be usefull in daily practice in ICU

Author Response

Dear reviewer,

thank you for reviewing our article. your kind remarks will mean a lot to us in the continuation of our work in the field of atrial fibrillation and pulmonary embolism. 

Ivica Djuric

Reviewer 2 Report

Thank you for giving me the chance to review this paper. Djuric et al. performed this study to determine the impact of AF type (paroxysmal AF compared to persistent/permanent AF) on sex-related short-term all-cause mortality in patients with acute pulmonary embolism. In this background, the authors retrospectively analyzed overall 1600 patients (743 males and 857 females) with acute PE. They concluded that the occurrence of paroxysmal AF in female patients with acute PE has predictive value for all-cause hospital mortality independent of age and mortality risk.

1. Figure 1 is broken, and needs to be improved.     

2. Table 1. It will be good to present a p-value comparing male and female patients. There is a big difference between the two groups in age.  

3. Primary and secondary outcomes can be defined.

4. The impact of the type of AF on the whole population can be presented.   

Author Response

Dear reviewer,

thank you for reviewing our article. your kind remarks will mean a lot to us in the continuation of our work in the field of atrial fibrillation and pulmonary embolism. 

  1. In our version of the manuscript Figure 1. does not appear broken. It may be problem in different formating.
  2. In Table 1. p value regarding age difference is ≤0,001, and the value is entered in Table 1. again there may be problem with formating.  
  3. The primary end-point of the study was all-cause hospital death. The secondary end-point of the study is that  adding paroxysmal AF to the mortality risk model in female patients can increase significantly its prognostic power.
  4. There are publication regarding the impact of type of AF on the general population and some of these are listed in thereferences. Our goal was to highlight the gender difference in relation to the onset of AF. 

Ivica Djuric

Reviewer 3 Report

The aim of the authors was to assess to prognostic significance of acute/chronic AF in patients with acute PE 2. It is not particularly original nor particularly relevant.

The authors have studied a very large population. Therefore, although there is no specific a priori hypothesis to be tested, their results could be useful to stimulate further research on this topic.

I have no suggestions to improve the methodology.

The conclusions are consistent. The limit is that there is no clear and specific hypothesis to be tested.

appropriate references

no further comments on the tables and figures

I would only ask the authors to have the text carefully reviewed by a native English speaker

Author Response

Dear reviewer,

thank you for reviewing our article. your kind remarks will mean a lot to us in the continuation of our work in the field of atrial fibrillation and pulmonary embolism. We are awere that our manuscript needs extensive revision of English Language so we will send it to editing service.  

Ivica Djuric